# FreeLM: Fine-Tuning-Free Language Model

## Abstract

Pre-trained language models (PLMs) have achieved remarkable success in NLP tasks. Despite the great success, mainstream solutions largely follow the *pre-training then fine-tuning* paradigm, which brings in both high deployment costs and low training efficiency. Nevertheless, fine-tuning on a specific task is essential because PLMs are only pre-trained with *language signal* from large raw data. In this paper, we propose a novel *fine-tuning-free* strategy for language models, to consider both *language signal* and *teacher signal*. *Teacher signal* is an abstraction of a battery of downstream tasks, provided in a unified proposition format. Trained with both language and strong task-aware teacher signals in an interactive manner, our FreeLM model demonstrates strong generalization and robustness. FreeLM outperforms large models *e.g.,* GPT-3 and InstructGPT, on a range of language understanding tasks in experiments. FreeLM is much smaller with 0.3B parameters, compared to 175B in these models.

## 1 Introduction

Pre-trained language models (PLMs), exampled by BERT Devlin et al. (2019), the GPT series Radford et al. (2018; 2019); Brown et al. (2020) and their variants Liu et al. (2019); Joshi et al. (2020); Sun et al. (2019); Raffel et al. (2020), have been widely applied in various language processing tasks and achieved remarkable success Zhao et al. (2023).

Despite their great success, the *pre-training then fine-tuning* paradigm Devlin et al. (2019); Radford et al. (2019) brings in very large costs for training and deployment in enterprise-level applications. Even large companies are very careful in using billion-parameter PLMs online Sanh et al. (2019), and remain showing high interest in small models. Before the era of PLMs, small models are trained for specific tasks. For some tasks, these task-dedicated models may perform comparably to or even better than large PLMs with fine-tuning Grinsztajn et al. (2022). Both the training and deployment costs are manageable since the task-specific datasets and models are typically much smaller. Nevertheless, additional efforts are essential to design task-specific models, one for each task. Illustrated in Figure 1, pre-trained with much larger datasets, PLMs gain high generalization, and can be fine-tuned to diverse downstream tasks for language understanding and generation. The effort of designing task-specific small models is now replaced by the huge costs of pre-training on large datasets and the high deployment costs.

To reduce deployment costs, zero-shot *e.g.,* GPT-3 Brown et al. (2020) and few-shot models have been investigated. Their performance, particularly on understanding tasks, remains unsatisfactory. One of the reasons is that the PLMs are trained with only *language signal* which is not task-aware, *i.e.,* the training objective of PLMs does not well align with the task objective. Recently, instruction-tuning-based models, *i.e.,* InstructGPT Ouyang et al. (2022) and FLAN Wei et al. (2022), further improves zero-shot performance. Their main design transfers everything into language text, then uses self-supervised *language signal* to train the model. In this way, the *language signal* in their training becomes more task-aware and relatively stronger. However, to achieve the best performance on specific tasks, adaptation remains necessary.

The idea of enhancing task awareness in training motivates us to consider a *reasonably sized model* that is capable of generalizing to *a good range of pre-defined tasks*. The ultimate goal is to achieve good performance on all these pre-defined tasks (and potentially unseen tasks) with low training and deployment costs. We believe this is an ideal setting for an enterprise to handle all their own specific tasks with a single model trained on their own data, without additional fine-tuning.

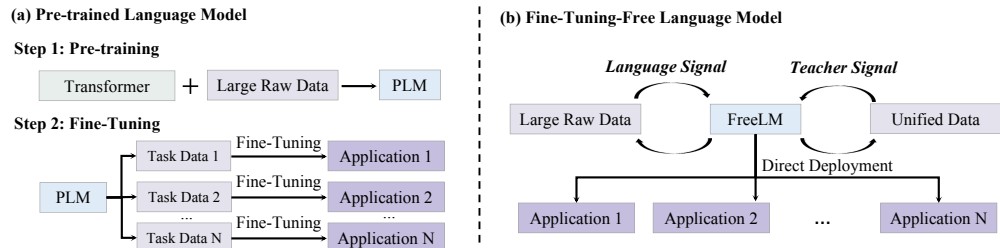

Figure 1: An overview of pre-trained language models and our proposed fine-tuning-free language model.

The key challenge then becomes how to unify the pre-defined tasks as a strong task-aware supervision signal for model training. To this end, we propose FreeLM, a fine-tuning-free language model. Illustrated in Figure 1, FreeLM is trained with both *language signal* and *teacher signal* in an iterative way. During training, in different iterations, the model alternately learns (i) the *language signal* from large raw data, and (ii) the *teacher signal* from unified data prepared based on all pre-defined tasks. Benefiting from the interplay between both signals, FreeLM is able to handle all pre-defined tasks with high accuracy, and handle unseen tasks well as a typical pre-trained language model does.

In addition to the *fine-tuning-free* strategy, a major contribution is that we make the very first attempt to unify a good range of language tasks through a unified data format, called *proposition format*. Datasets constructed for all pre-defined tasks are then mapped to this unified proposition format, and become the unified data in Figure 1. Through the proposition format, we provide proposition correctness judgment to help the language model know the facts. The unified format also implicitly facilitates mutual improvements among these tasks, which further contributes to better generalization and robustness.

To the best of our knowledge, this is the first attempt to propose an effective *fine-tuning-free* strategy for large language model training. We also demonstrate how to unify multiple tasks through a unified format to make the PLM more task-aware through strong supervision signals. Through experiments, we show that FreeLM greatly outperforms state-of-the-art models on language understanding tasks, compared to GPT-2, 175B GPT-3 and InstructGPT. We also show that the generation ability of FreeLM is even better than GPT-2. FreeLM achieves good performance on unseen tasks, and has excellent generalization and robustness.

## 2 RELATED WORK

In this work, we focus on both language generation and language understanding tasks. Thus, we only consider unidirectional Transformers like GPT as basic models, and do not consider masked language models (*e.g.,* BERT) or encoder-decoder language models (*e.g.,* T5). Next, we brief auto-regressive language models and instruct-tuning-based models.

### 2.1 AUTO-REGRESSIVE LANGUAGE MODELS

Auto-regressive language models are trained to predict the next token based on all previous tokens. The unidirectional nature empowers these models with language generation capability. The impacts of model scale and model structure are the two key research directions.

On model scale, the most typical and also the most influential auto-regressive LMs are the GPT series, *i.e.,* GPT-1 Radford et al. (2018), GPT-2 Radford et al. (2019), and GPT-3 Brown et al. (2020). In particular, the success of GPT-3 has made researchers realize that the violent aesthetics of the model scale and large raw data can have such a good generation performance. To make the models bigger, even methods like MoE Nie et al. (2021) are proposed. However, large scale brings in large costs and many challenges, particularly the high costs of fine-tuning on downstream tasks.

To improve the model structure, GLM Du et al. (2022) is designed to utilize the autoregressive blank infilling. Transformer-XL Dai et al. (2019) tries to solve the longer-term dependency, and its variant XL-Net Yang et al. (2019) hopes to learn bidirectional contexts. However, these models do not

perform better than GPT-3 as the models get larger. The vital reason remains how to effectively and efficiently utilize data.

## 2.2 INSTRUCT-TUNING BASED MODELS

A larger model size does not mean that it can produce output that better meets user expectations Ouyang et al. (2022). One solution is to fine-tune large language models based on a wide range of tasks, human feedback, and so on. The typical and most representative model is Instruct-GPT Ouyang et al. (2022). Through reinforcement learning, data from tasks, and human feedback, InstructGPT achieves impressive performance on both language understanding and generation tasks. Experiments even show that InstructGPT can align with humans compared with original GPT-3. EFL Wang et al. (2021), designed for few-shot learning, reformulates potential NLP tasks into an entailment task based on RoBERTa-large. Entailment task is hard to handle well more types of tasks, while our unified proposition is more generic to fit more tasks. In addition, EFL suffers from degradation in generation abilities.

In our model design, we aim to achieve better results on both understanding and generation tasks without fine-tuning. InstructGPT adopts a method like "all in language". EFL gives up the generation ability. To our understanding, these models do not effectively utilize the *teacher supervision signal*. In this paper, we design an iterative training strategy on both language raw data and task-aware data. The key idea here is to teach the model to be task-aware data, while not forgetting its role of a language model, *i.e.,* to model language.

## 3 TASK UNIFICATION

Our goal is to train a task-aware language model which learns from language as a typical PLM does, and also learns from a good number of task-specific datasets. Shown in Figure 1, FreeLM learns from both language data and unified data, where the latter is the result of unifying pre-defined tasks.

**Language Data.** For language data, the choice is relatively straightforward. We adopt OpenWeb-Text Gokaslan & Cohen (2019), an open-source replication of the WebText Radford et al. (2019) dataset proposed by OpenAI. It is built by scraping all outbound links from the social media platform Reddit, which received at least 3 karmas. We use the same settings as GPT-2 on this dataset.

**Unified Data in Proposition Format.** Our key motivation is to enable enterprises to handle all their own specific tasks with a single model trained on their own data, without additional fine-tuning. However, it is challenging to access and conduct experiments on proprietary data. Even if that is feasible, our results will be hard to be benchmarked against the current state-of-the-art. Without loss of generality, we choose to unify seven well-defined, popular, and representative language tasks: 1. *question answering*, 2. *paraphrasing*, 3. *topic classification*, 4. *story cloze*, 5. *sentiment classification*, 6. *natural language inference*, and 7. *linguistic acceptability*.

We unify these seven tasks by transforming them into a "proposition correctness judgment" task, to judge whether a proposition is true. Specifically, for each task, we design some candidate templates, then use the designed rule to transfer each instance into a ***proposition format***:

> "[tsk] *task name* [tsk] *instance text(s)* [sep] *dynamic prompt* [cls]"

The *task name* follows the original pre-define task name before unification like "question answering". Because each task has its own characteristic, hence we use *dynamic prompt* to fit them well considering the ground-truth label.

We take the *topic classification* task as an example to explain the mapping, also illustrated in Figure 2. We convert the class label into a natural language description based on a template selected from a pre-designed template pool. In Figure 2, Template 1 is chosen among the $N$ templates. Then we could build a candidate proposition statement by appending the natural language description to the original input text. The natural language description is the class labels in this example task.

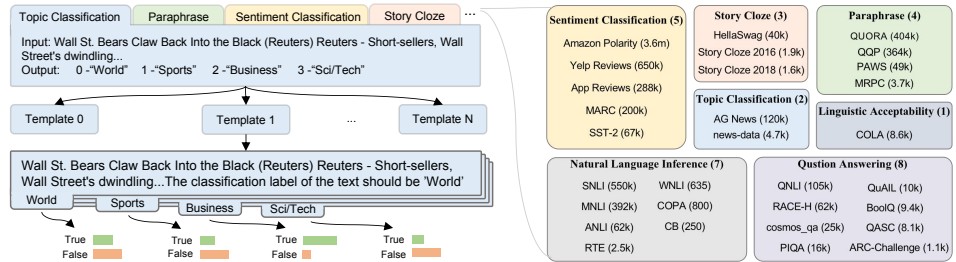

Figure 2: A template example for the topic classification task. This example also describes the transform rule for task unification. The right of this figure details the list of 30 datasets for 7 tasks, and numbers in parentheses indicate dataset size.

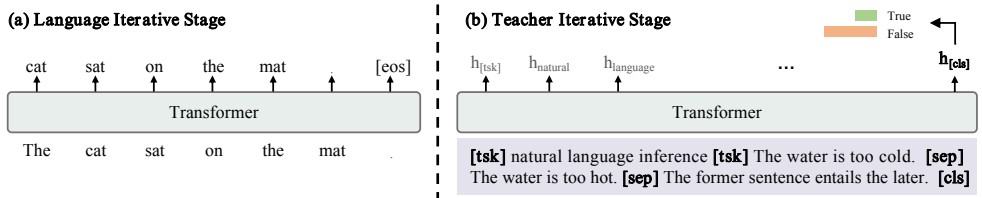

Figure 3: (a) The language stage utilizes the standard language training method, and (b) the teacher stage uses the unified proposition signal.

In our implementation, we utilize 30 datasets of the seven chosen tasks (see the right part of Figure 2). For each dataset, we design at most 10 templates to build the proposition. Each item will randomly choose one template when building unified data. According to the chosen template, the original input context is concatenated with the label description to form a proposition. With the unified formatted proposition as input, our FreeLM is responsible for giving a true-or-false judgment.

FreeLM designs task unification by unified proposition format, which brings a good generalization effect. In other words, FreeLM only has one proposition correctness judgment task (except language task), which endows FreeLM with the ability to handle almost all NLU tasks. As a comparison, multi-task needs special design for each task to fit the multi-task model. More importantly, multi-task could hardly gain generalization ability except the designed tasks.

## 4 FREELM

### 4.1 ARCHITECTURE AND ITERATIVE TRAINING

Figure 3 shows the architecture of the proposed FreeLM. The left part of the figure shows training with language data, similar to the mainstream generative language models. The right part shows training with task-aware unified data, through making correct judgments on unified propositions.

To train the model with both language and unified data, a straightforward choice is to first train the model fully on language data, then fully on unified data, similar to pre-train then fine-tuning. However, this strategy would lead the model to focus too much on tasks, and reduce its language capability and generalization. In our design, *language signal* and *teacher signal* are used for training in an iterative manner, analogous to "first read, then exercise". Next, we detail the two training stages, which take turns to train the model.

**Language Iterative Stage.** To keep the generation ability of language models, we choose the auto-regressive language model, more specifically GPT-2, as our base model in FreeLM. As shown in Figure 3(a), *language signal* guides the model to predict the next token with the previous token(s). Given a text consists of $N$ tokens $\mathcal{T} = \{t_1, \ldots, t_N\}$, FreeLM computes the probability $P_L$ of token $t_k$ with the previous $k-1$ tokens:

$$P_L(t_k|t_1, \ldots, t_{k-1}) = \text{softmax}(W_v h_{t_{k-1}}) \tag{1}$$

Here, $h_{t_{k-1}}$ denotes the representation encoded by Transformer with the previous tokens $\{t_1, \ldots, t_{k-1}\}$ as input. $W_v$ represents the learnable parameters.

**Teacher Iterative Stage.**   In the proposed fine-tuning-free structure, *teacher signal* aims to guide the model to learn task-oriented knowledge. After unifying all tasks to proposition correctness judgment, the model is expected to make correct judgements about the propositions. Let $u = (T_p, y)$ be a pair where $T_p$ refers to the proposition statement, and $y$ is the label of this proposition. When the proposition is right, $y$ is True, otherwise False.

Figure 3(b) shows the mechanism of *teacher signal* from unified data. According to the designed transfer rule (see Section 3), each instance in the unified data is appended with a dedicated token "[cls]". Given an instance, we can calculate the representation $h_{[cls]}$ of token "[cls]" by the shared Transformer of the language model. The proposition distribution $\mathcal{P}_P$ is computed by a softmax classifier based on the learned representation $h_{[cls]}$:

$$\mathcal{P}_P(y|T_p) = \text{softmax}(W_p h_{[cls]} + b_p), \tag{2}$$

where $W_p$ and $b_p$ are learnable parameters.

## 4.2   TRAINING OBJECTIVE

The training objective design has two parts. One is to maximize the likelihood of predicted tokens, defined in Equation 3. The other is to minimize the cross-entropy of judging proposition correctness, defined in Equation 4, where $y$ is the ground truth of the given $T_p$.

$$J(\theta) = \sum \sum_k \log P_L(t_k|t_1, \ldots, t_{k-1}) \tag{3}$$

$$U(\theta) = \sum \text{cross-entropy}\left(\mathcal{P}_P(y|T_p), y\right) \tag{4}$$

Note that the two objectives are alternately performed in each iteration, along with the two types of training data.

## 5   EXPERIMENTS

We evaluate FreeLM from two perspectives: language understanding performance, and language generation performance. Then we perform a detailed analysis of the impact of iterative processes, the impact of special tokens in the proposition format, and the generalization ability on unseen data.

## 5.1   TRAINING DATASETS

As indicated in Section 3, for raw language data, we adopt the OpenWebText Gokaslan & Cohen (2019). The unified data are composed with 30 datasets, listed in Figure 2, from the seven tasks. Training sets are collected for all these datasets, except Story Cloze 2016 & 2018, whose validation sets are collected.

For each instance in unified data, we construct a positive sample (*i.e.,* true proposition) with the gold label according to the build method. Simultaneously, we randomly choose another label to construct a false proposition as a negative sample. In training set building, only one template would be randomly chosen for each original instance. The maximum number of training set proposition samples per dataset is set to $100K$. The final unified training dataset contains about $1.76M$ samples.

## 5.2   EVALUATION ON LANGUAGE UNDERSTANDING

A key motivation of FreeLM is the understanding ability without fine-tuning. For understanding ability, we evaluate FreeLM against strong baselines, including 175B GPT-3, InstructGPT,[1] and GPT-2, under the settings of zero-shot and few-shot on these baseline models.

---

[1]We use text-davinci-003 version through OpenAI API.

Table 1: Comparison with GPT-2, GPT-3 and InstructGPT on GLUE validation set. *F.T.*, *Z.S.* and *F.S.* denote fine-tuning, zero-shot and few-shot, repectively. Best results are in bold face (without considering GPT-2 w. F.T.).

| Model | CoLA Mat. Corr | SST-2 Acc. | MRPC F1 | MRPC Acc. | QQP F1 | QQP Acc. | MNLI Mat. Acc. | MNLI M.m. Acc. | QNLI Acc. | RTE Acc. | Average – |
|---|---|---|---|---|---|---|---|---|---|---|---|
| | | | | | *Fine-Tuning* | | | | | | |
| GPT-2 *w. F.T.* | 56.52 | 94.95 | 87.93 | 82.84 | 87.91 | 90.83 | 85.87 | 85.83 | 91.03 | 68.23 | 83.19 |
| | | | | | *Few-Shot & Zero-Shot* | | | | | | |
| GPT-2 *w. Z.S.* | -3.26 | 51.83 | 81.22 | 68.38 | 52.88 | 36.19 | 35.60 | 34.30 | 46.40 | 51.62 | 45.52 |
| GPT-2 *w. F.S.* | 0.60 | 63.30 | 81.34 | 68.62 | 44.46 | 46.80 | 31.60 | 35.00 | 48.60 | 49.09 | 46.94 |
| GPT-3 *w. Z.S.* | 17.28 | 52.98 | 65.05 | 53.92 | 4.61 | 62.80 | 36.80 | 39.50 | 60.90 | 64.98 | 45.88 |
| GPT-3 *w. F.S.* | -4.45 | 93.34 | 70.92 | 59.80 | 6.31 | 64.40 | 44.10 | 46.60 | 53.80 | 65.70 | 50.05 |
| InstructGPT *w. Z.S.* | **63.81** | 92.08 | 83.75 | 77.94 | 64.20 | 78.70 | 46.20 | 48.90 | 71.89 | **86.64** | 71.41 |
| InstructGPT *w. F.S.* | 56.72 | **95.18** | 77.41 | 70.83 | 72.98 | 81.20 | 71.70 | 72.89 | 81.30 | 82.67 | 76.29 |
| FreeLM | 53.00 | 93.81 | **87.21** | **82.11** | **85.47** | **89.60** | **81.80** | **79.80** | **88.30** | 71.12 | **81.22** |

We mainly evaluate FreeLM in *fine-tuning-free* scenarios. That is, our FreeLM does not conduct further task-oriented fine-tuning after training. Also, as an upper bound on the understanding ability with the same scale, we show the results of GPT-2 fine-tuning.

**Evaluation Metric.** We choose General Language Understanding Evaluation (GLUE) as the benchmark, which consists of typical natural language understanding tasks, including *natural language inference* (*i.e.,* AX, MNLI Williams et al. (2018), RTE Bentivogli et al. (2009), QNLI Wang et al. (2018), and WNLI Levesque et al. (2012)), *sentiment classification* (*i.e.,* SST-2 Socher et al. (2013) and STS-B Cer et al. (2017)), *paraphrase* (*i.e.,* MRPC Dolan & Brockett (2005) and QQP) and *linguistic acceptability* (*i.e.,* CoLA). Validation sets of each dataset are used for evaluation. WNLI is excluded from the benchmark following the previous work Devlin et al. (2019) because the GLUE webpage[2] notes that there are issues with the construction of this dataset. In addition, Ax is excluded due to the lack of the validation set. As the regression task (*i.e.,* STS-B) is not suitable for FreeLM, we also exclude it from the benchmark.

We use the remaining seven tasks from the GLUE benchmark to evaluate. For experimental efficiency and cost effectiveness on API, when evaluating GPT-3 and InstructGPT, we use the validation set of GLUE for evaluation, and sample $1k$ instances for each task as the final evaluation set. All results are compared fairly in the same setting.

**Results against Baselines.** The GLUE results are reported in Table 1. On both the average metric and 7 out of 10 evaluation metrics, FreeLM outperforms all baselines, revealing its strong understanding ability. In particular, our FreeLM gains about $5$ points improvement on average performance compared with the powerful InstructGPT. We also observe that MRPC, QQP, MNLI and QNLI tasks obtain 7 or more points improvement in absolute accuracy. More importantly, the scale of our FreeLM is only 0.3B. Even compared to the much larger GPT-3 with 175B parameters, FreeLM achieves the state-of-the-art in the downstream tasks without fine-tuning, thanks to the strong task-aware signal in training. Among baseline models, InstructGPT has significant advantages over GPT-2 and GPT-3. Our results also show that InstructGPT is an extremely strong model.

As reported in Table 1, our FreeLM nearly matches the performance of *GPT-2 with fine-tuning*. As GPT-2 is fine-tuned individually on each dataset, the comparison in essence is between one FreeLM and 7 dedicated GPT-2(s). In this sense, GPT-2 with fine-tuning could represent an approximate upper bound on 0.3B scale. InstructGPT, the strongest generative PLM, achieves the second-best performance. As a comparison, FreeLM uses a classification-based approach while instruction tuning uses a generation-based approach, which results in our model being more effective. More importantly, the task unification (see Section 3) by one unified proposition format brings a good generalization effect. An interesting point is that FreeLM performs significantly worse than InstructGPT, but significantly better than GPT-2 with fine-tuning, on the accuracy of RTE. This result also suggests that the ability of language models could be positively related to the scale.

---

[2] https://gluebenchmark.com/faq

Table 2: Perplexity on FreeLM and GPT-2.

| Model | LAMBADA | WikiText2 | PTB |
|-------|---------|-----------|------|
| FreeLM | 38.44 | 33.66 | 28.47 |
| GPT-2 | 35.31 | 30.23 | 26.62 |

Table 3: Case study on generation ability. Consistency and inconsistency expressions are in blue and red colors.

| *Case A* |
|---|
| **INPUT:** Rick grew up in a troubled household. He never found good support in family, and turned to gangs. **It wasn't long before Rick got shot in a robbery. The incident caused him to turn a new leaf.** |

| **GPT-2:** "I was a gang member, but I didn't know it at the time," Rick said. "I had no idea what was going on. I just wanted to get away from it. That's how I ended up on the streets. And that's why I'm here today." | **FreeLM:** "When I got out of prison, I was a different man," he says. "I was able to get my life back on track. I went back to school, got a job, started a family. And that's what I've been doing ever since." |
|---|---|

| *Case B* |
|---|
| **INPUT:** Tony was happy to be going on his first cruise. He arrived at the docks and was ready with all his luggage. When he walked around the corner he couldn't believe the size of it. **He boarded and the ship left the docks.** |

| **GPT-2:** When he got to his room he was surprised to see that the room was bigger than he thought it would be. There was a bed, a desk, and a TV. ... | **FreeLM:** "This is the first time I've ever been on a cruise ship," he said. "I've never seen anything like this before." |
|---|---|

| *Case C* |
|---|
| **INPUT:** Jasper has been avoiding getting a real job for months. He has hated all the previous office jobs that he had. One day **he sees a job opening at his local butcher**. He applies successfully for the job at the butcher. |

| **GPT-2:** "He said, 'I don't want to hire you because you're too old,'" Jasper says. "And I was like, I'm not old. I've been doing this for a long time, and I know what it's like to be rejected. ... | **FreeLM:** "It's a great job," he says. "I love it. I love the people here. They treat me like a king. The people are so nice. It makes me feel like I can do anything I want to do in this world. ... |
|---|---|

Not directly reflected in Table 1, through experiments, we also notice that FreeLM is robust and insensitive for inference. Interestingly, we find that the results of the GPT family models predicted by few-shot or zero-shot are sensitive to the settings of templates, parameters, and post-processing methods. For example, we measure GPT-2 on the MRPC dataset with a zero-shot setting. The results of the vocabulary probability distribution $top\_k = 50$ are used to predict the classification. If we calculate the average sum of the probability ratios of positive and negative words, the accuracy is $33.08$. However, if we get the word by the highest probability value, the accuracy is more than doubled to $68.13$. The gap is so large that downstream tasks are hard to use reliably. On the other hand, FreeLM is robust without the need of expert knowledge or parameter tuning.

## 5.3 EVALUATION ON LANGUAGE GENERATION

As aforementioned, while FreeLM gains better understanding ability, we hope that the model retains its language generation ability, similar to GPT-2, for learning from the same language data. Hence, we evaluate the generation performance of FreeLM against GPT-2. The evaluation is conducted on three public datasets, *i.e.,* LAMBADA Paperno et al. (2016), WikiText2, and PTB. We use perplexity (PPL) to evaluate the generation performance, following common practice.

**Results against Baselines.** Table 2 reports the perplexity (PPL) of FreeLM and GPT-2. The PPL of FreeLM is slightly higher than GPT-2. The minor gap could reflect that our model nearly reaches GPT-2 in terms of generation. There are studies suggesting that PPL does not fully reflect the generation ability of language models Wang et al. (2022). Nevertheless, there are no good alternative metrics.

Table 4: The effect of *language signal* for understanding, where *L.S.* refers to *language signal*.

| Model | CoLA Mat. Corr | SST-2 Acc. | MRPC F1 | MRPC Acc. | QQP F1 | QQP Acc. | MNLI Mat. Acc | MNLI M.m. Acc | QNLI Acc. | RTE Acc. | Average – |
|---|---|---|---|---|---|---|---|---|---|---|---|
| FreeLM | **52.53** | **93.81** | **87.21** | **82.11** | **86.32** | **89.82** | **80.70** | **80.93** | **88.61** | 71.12 | **81.32** |
| *w.o. L.S.* | 40.14 | 92.43 | 85.71 | 79.17 | 85.66 | 89.11 | 77.56 | 78.57 | 87.28 | **71.48** | 78.71 |

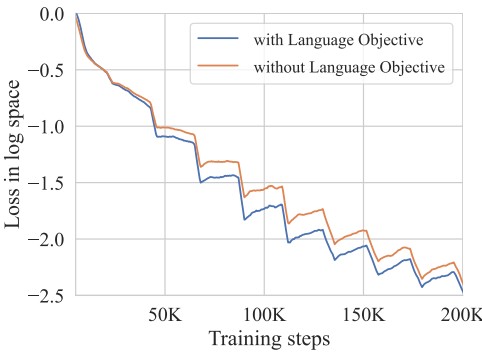

Figure 4: Effect of *language signal* on proposition loss.

Since generation evaluation does not have a gold standard, we provide a case study on a story cloze task. The input texts come from the StoryCloze dataset Mostafazadeh et al. (2016). In order to make the comparison objective and fair, we train a new FreeLM model by removing this dataset from the unified data. Top-k random sampling with $k = 50$ is used for generation, width of $5$ for beam search, and the maximum generated length is $256$. Table 3 shows three generated samples.

First, we find that the readability of the text generated by FreeLM is good by manual sampling inspection. There are no obvious grammatical errors, which shows that FreeLM and GPT-2 perform at the same level in the grammatical aspect. Second, Table 3 shows that there are many context inconsistency phenomena in the texts generated by GPT-2. Our FreeLM could generate more consistent text with the same texts as input. To clearly show the correct logical association, we use the text in blue to highlight the relation. Meanwhile, we use the text in red to warn the logical errors. Cases A and B show that our FreeLM is capable of generating more consistent texts. Interestingly, although there is no obvious grammatical error in Case B, the generated text by GPT-2 almost has no direct logical relationship with the input text. That is, it is not a high-quality generated result.

In simple words, our experiments show that the proposed FreeLM achieves the same level of PPL as GPT-2. FreeLM could generate more consistent text. We believe that if we train our model at a larger scale, better generation ability can be achieved.

## 5.4 DETAILED ANALYSIS

We conduct three sets of experiments to study (i) the impact of the iterative training, (ii) the proposition format, and (iii) the generalization ability on unseen tasks.

**The Iterative Training.** If we remove the *teacher signal*, FreeLM will degenerate into a general language model. Based on the results in Section 5.2 and 5.3, we conclude that the discarding of the *teacher signal* will lead to a reduction in the performance of understanding tasks. At the opposite extreme, we remove the *language signal*. The model could then only rely on the objective of proposition correctness judgment for training. The comparison is under the same training parameters and the same number of steps. Table 4 shows the results compared to standard FreeLM. After removing *language signal*, almost all evaluation metrics show a significant drop. Interestingly, the PPL of WikiText2 could increase to $8,000$, even $9,000$, compared to a regular scale 30 from GPT-2. As a result, the generation ability is damaged seriously.

We have shown that *language signal* could help FreeLM learn better. Also, to evaluate whether FreeLM can learn faster, we give the loss performance during training in Figure 4. We observe that

Table 5: The effect of task prefix for understanding.

| Model | CoLA Mat. Corr | SST-2 Acc. | MRPC F1 | MRPC Acc. | QQP F1 | QQP Acc. | MNLI Mat. Acc | MNLI M.m. Acc | QNLI Acc. | RTE Acc. | Average – |
|---|---|---|---|---|---|---|---|---|---|---|---|
| FreeLM | **52.53** | **93.81** | **87.21** | **82.11** | **86.32** | **89.82** | **80.70** | **80.93** | 88.61 | 71.12 | **81.32** |
| *w.o. prefix* | 48.15 | 92.78 | 86.76 | 81.37 | 86.06 | 89.53 | 80.23 | 80.32 | **88.76** | **71.84** | 80.58 |

Table 6: FreeLM vs. GPT-2 fine-tuning on unseen data.

| Model | MRPC F1 | MRPC Acc. | RTE Acc. | COPA Acc. | CB Acc. | Average – |
|---|---|---|---|---|---|---|
| FreeLM | 87.21 | 82.11 | 71.12 | 73.00 | 73.21 | 77.33 |
| FreeLM$_U$ | 81.22 | 72.06 | 66.79 | 68.00 | 68.64 | 71.34 |
| GPT-2 *w. F.T.* | 87.93 | 82.84 | 68.23 | 54.00 | 73.21 | 73.24 |

after $20k$ steps, the proposition loss of FreeLM decreases faster. It shows that the *language signal* has a positive effect on task-oriented learning. In conclusion, *language signal* and *teacher signal* could promote each other under the iterative training method.

**The Proposition Format.** Task prefix, such as "[tsk] Topic Classification [tsk]", could guide the model to narrow down the search space. In the implementation, the set of task prefix is {*Linguistic Acceptability*, *Topic Classification*, *Story Cloze*, *Sentiment Classification*, *Question Answering*, *Paraphrase*, *Natural Language Inference*}. Since the parameters of FreeLM are small, we utilize this kind of task prefix to guide the model when meeting task-oriented understanding. To compare, we train another FreeLM without task prefix. Other settings are the same as FreeLM. Table 5 gives the comparison results.

We observe that the performance of most tasks is at a similar level. This phenomenon could prove that FreeLM does not significantly rely on task prefix. We believe that when the size of FreeLM exceeds $1B$, task prefix will have little effect.

**Generalization to Unseen Dataset.** To evaluate the generalization ability of FreeLM, we train a new model FreeLM$_U$ by removing 4 datasets from unified data. They are MRPC, RTE, COPA Gordon et al. (2012) and CB De Marneffe et al. (2019). Other settings are exactly the same. Table 6 gives the results compared to standard FreeLM.

We observe that the average score of FreeLM$_U$ nearly matches the performance of *GPT-2 with fine-tuning*. More interestingly, the accuracy of the FreeLM$_U$ on the COPA dataset has increased by $14$ points than *GPT-2 with fine-tuning*. Compared to the standard FreeLM, the average drop is about 6 points. It is an inspiring result because it reveals that our FreeLM has a good generalization ability. We believe the reason is that we unify various tasks into one proposition correctness judgment task. The phenomenon also reminds us that the design of *teacher signal* is a key to training a better language model.

## 6 CONCLUSION

With the aim of reducing costs in training and deployment, we design a novel fine-tuning-free language model. The model training benefits from the self-supervised *language signal* as a typical language model does. It also becomes task-aware through the training on unified data. Evaluations show that FreeLM retains its strong language capability as a language model in a similar scale *i.e.,* GPT-2. Experiments also show that FreeLM's understanding performance on the pre-defined tasks is significantly better than GPT-3 and instructGPT with zero- or few-shot settings, even though our model is much smaller, 0.3B versus 175B parameters. FreeLM also generalizes well if a task can also be mapped to the unified format. We believe the key reason behind the strong performance of FreeLM is the task unification and the alternative training. If an enterprise can design a similar unified data format for all their tasks, like the proposition format in our setting, the proposed fine-tuning-free training strategy could be a good consideration for cost saving.

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

# A APPENDIX

### IMPLEMENTATION DETAIL

We list the implementation details of the proposed FreeLM to support reproduction. To support research, we will release the corresponding codes and trained parameters upon acceptance.

In our experiment of training FreeLM, we choose the medium version of GPT-2 with 345M parameters, which makes up of 12 transformer blocks. The model dimension is 768. To speed up training and save energy, we initialize the model with the pre-trained parameters provided by HuggingFace. All the experiments are performed on 5 nodes each with eight 40GB NVIDIA Tesla A100 (40 cards total) for about 8 days. We train FreeLM for 8 epochs on OpenWebText with a batch size of 4 on each card. Each epoch takes around 13.5 to 17.5 hours to complete. We use AdamW optimizer with a learning rate of 5e-5. The mixing ratio of the language iterative stage and teacher iterative stage samples is 2:1. Experiment results on various mixing ratios show that 2:1 performs better on both NLU and NLG tasks. A more precise ratio would lead to significantly high computational cost. On the other side, enhancement on the raw data part benefits both NLU and NLG tasks due to the sharing transformer blocks. Hence, more samples from raw data are necessary.

COMPUTATION COST

In the training process, we set the context window size to 256 for teacher iterative stage, which is 1/4 of the context window size (*i.e.,* 1024) for language iterative stage. Additionally, by using a 2:1 mixing ratio of language iterative stage and teacher iterative stage samples, the additional computation cost for FreeLM's training is reduced to less than 1/8.

