# OpenReview forum: "FreeLM: Fine-Tuning-Free Language Model"
_ICLR.cc/2024/Conference — Submitted to ICLR 2024_

### Official Review · Reviewer_hgDz · 2023-10-28

**Soundness:** 2 fair
**Presentation:** 3 good
**Contribution:** 2 fair
**Rating:** 3
**Confidence:** 3

**Summary:**

The paper propose FreeLM as a finetuning-free method for downstream tasks. Compared with pretraining then fine-tuning paradigm, FreeLM fuses the task data into pretraining data and is iteratively trained in the pretraining stage. On several classification tasks, FreeLM is shown to be better than much larger GPT-3.

**Strengths:**

- Injecting task bias into pretraining stage is an interesting idea.
- The paper is mostly well written, with intensive experiments and analysis on various aspects of the proposed method.

**Weaknesses:**

- **Need more in-depth analysis regarding the influence on NLG tasks**: As the author mentioned that FreeLM has a higher perplexity than GPT-2, it would be more informative to provide a numerical comparison on NLG benchmarks instead of only showing a few cases.
- **Missing comparison with multitask fine-tuning**: Table 4 does not include the results of the multitask fine-tuning (https://openreview.net/forum?id=gEZrGCozdqR) baseline, which is also learning-free when applied to other tasks. It would be important to compare with multitask fine-tuning results and analyze the generalization ability compared with FreeLM.
- **Scalability**: In-context learning is another learning-free method when performing downstream tasks and the performance of which gains a lot after scaling. However, there is a lack of comparison with in-context learning when model scales up.

**Questions:**

- Can FreeLM benefit broader tasks (e.g., language generation, code generation) beyond binary classification?
- How about the performance and generalization ability compared with multitask fine-tuning?
- How about the FreeLM performance compared with in-context learning after model scaling?

**Details Of Ethics Concerns:**

None.

---

### Official Review · Reviewer_NrTr · 2023-10-28

**Soundness:** 1 poor
**Presentation:** 1 poor
**Contribution:** 2 fair
**Rating:** 1
**Confidence:** 5

**Summary:**

In this paper, the authors introduce a novel approach to pretrain a decode-only language model, eliminating the need for subsequent fine-tuning. Drawing on prompt-based learning principles, their model, FreeLM, excels in a variety of downstream tasks. Impressively, it surpasses the performance of larger models like GPT-3 and InstructGPT on the GLUE benchmark, despite having substantially fewer parameters.

**Strengths:**

1. By merging pretraining and finetuning into one unified stage, the authors present a potentially more efficient training approach.

2. Remarkably, despite its smaller size, FreeLM surpasses larger models like GPT-3 and InstructGPT in GLUE's downstream tasks.

**Weaknesses:**

1. While the "teacher iterative" method is an interesting take on prompt-based learning, merging pretraining and prompt-finetuning into a single step may not be as novel as it seems. A clear distinction and justification for this fusion could strengthen the paper.

2. The choice of the GLUE benchmark for evaluation does not make sense to me for a few reasons:

    1. GLUE is typically used to assess single-task, supervised learning models, mainly encoder-only (BERT-like) or sometimes encoder-decoder LMs (T5-like). However, decoder-only LMs, like FreeLM, may not perform as well on these tasks. For instance, decoder-only LMs generally score around 80 on MNLI, while state-of-the-art encoder-only models can achieve scores over 90. Even BERT-Base can achieve an accuracy of 84.

    2. The paper's approach to evaluating "generalization to unseen datasets" does not follow standard practices. A more established benchmark for zero-shot generalization of LMs on classification tasks is T0-Eval (https://arxiv.org/pdf/2110.08207.pdf), which includes 62 training datasets and 12 evaluation datasets.

    3. It is more appropriate to assess decoder-only LLMs on generative tasks, such as MathQA, HumanEval, MBPP, or OpenBookQA, rather than GLUE.

3. The comparison between FreeLM, GPT-3, and InstructGPT may not be entirely fair. A better approach would be to create a pretraining-finetuning baseline or to fine-tune an open-source LM like LLaMa on the GLUE benchmark for comparison.

4. Lastly, the paper's writing could be improved for clarity and readability. The language should be fluent and succinct, rather than coming across as if it was translated from another language.

**Questions:**

1. Can you please provide a clearer distinction between the proposed "teacher iterative" method and existing prompt-based learning methods? What specifically sets your approach apart, given that both methods seem to be simply a mix of pretraining and fine-tuning?

2. Why was GLUE chosen as the primary benchmark for evaluation, especially given that it is generally used for single-task, supervised learning models and not typically suited for decoder-only language models like FreeLM?

---

### Official Review · Reviewer_JFLv · 2023-10-29

**Soundness:** 1 poor
**Presentation:** 3 good
**Contribution:** 1 poor
**Rating:** 3
**Confidence:** 4

**Summary:**

The paper unifies the NLP tasks and proposes a fine-tuning-free (though not training-free) method for training a language model, claiming better performance than GPT-3 on various tasks (a point I view with skepticism).

**Strengths:**

- The paper unified the NLP tasks, which could be helpful for research to evaluate LLM (but I believe [lm-evaluation-harness](https://github.com/EleutherAI/lm-evaluation-harness) already did a good job now)
- I like the figures.

**Weaknesses:**

Minor Weakness:
- I take issue with the term "fine-tuning-free" for the language model, as it suggests a misconception that the proposed method requires no training. I acknowledge that the community lacks a clear definition of "fine-tuning," commonly understood as "continuous training" on a pre-trained model. Nonetheless, the proposed language model also undergoes continuous training based on GPT-2, contradicting the "fine-tuning-free" label. Furthermore, even if you are starting training from scratch, calling it a "fine-tuning-free" language model is misleading if it indeed involves parameter updates.

Major Weaknesses:
- The paper uses many terms to embellish the method, such as "teacher signal" and "language signal," but in reality, the approach is a "co-training" scheme that trains on both target tasks and unsupervised English data. The paper argues that "pre-training and fine-tuning is not an optimal paradigm" and suggests "training them together." I do not recognize the benefits here. Is it to prevent catastrophic forgetting? If that's the case, it requires verification within the paper. Overall, this paper lacks novelty and fails to demonstrate the effectiveness of the proposed method.
- I assume one of the motivations behind "co-training" is to enhance generation ability. However, the authors did not substantiate this with perplexity scores. Some examples are unconvincing. More evidence is needed to support it, and if other methods are considered unreliable, human evaluation is required.
- The paper asserts that their proposed LM achieved SOTA, surpassing GPT-3, which is not accurate. Even straightforward fine-tuning on these tasks with GPT-2 outperforms Free-LM. Would it then be plausible for the authors to assert that "fine-tuning GPT-2 on these target data achieves SOTA and is superior to GPT-3"? The authors should explicitly state the limitations in this section. By the way, the authors also need to show details how they design the prompt for GPT-3, an improper prompt also has big impact on the performance.

**Questions:**

The paper lacks novelty, fails to demonstrate the soundness of the proposed method, and makes overstatements in its evaluation, which also lacks details when they evaluate baselines. Hence, I strongly recommend rejecting this paper.

---

### Official Review · Reviewer_nxBg · 2023-11-01

**Soundness:** 1 poor
**Presentation:** 2 fair
**Contribution:** 1 poor
**Rating:** 1
**Confidence:** 4

**Summary:**

The paper studies a special method that mixes various supervised finetuning (SFT) data together with the pretraining data for language models. This special method requires fitting existing SFT data from various tasks into a new binary classification format, where the supervision can be added as an extra term in the loss function. With this approach the pretraining and modified finetuning is conducted in iterations.

The authors claimed in the abstract that their method is "finetuning free" and a small GPT2 sized model trained with this approach "out performs large models e.g. GPT-3 and InstructGPT".

**Strengths:**

The original motivation of this work is on the right general direction. It is important to study methods that can improve a general purpose LLM beyond what is possible with pretraining along, leveraging smaller but high quality datasets. Many fruitful methods has been proven along these lines, such as FLAN, Instruction finetuning, self-instruct, RLHF, etc.

The paper also studies whether the auto-regressive language modeling abilities has been impacted by adding SFT signals, and find that with their method this impact has been minimal. This is an encouraging sign.

The paper has set out to study cross-task generalizations, which is another important topic. They find some interesting generalization effect on COPA.

**Weaknesses:**

Although the paper has many good motivations, it also has serious flaws.

1) The central claim is miss leading. They claims that their method is "finetuning free" and they compare their results to zero-shot and few-shot results. However, this is not the case. In particular, they still require gradient steps on the training split data of their eval tasks (or of a similar eval task). This is effectively finetuning, albeit with a slightly different loss function. In particular, using their method does not alleviate the user from collecting the finetuning set, which is importantly not the case for typical zero-shot and few-shot settings with a foundation LLM. Thus the claims involving comparison to GPT-3 and InstructGPT are unfair because those models did not explicitly have access to those finetuning training sets.

2) The proposed method lacks novelty. It is essentially doing multi-task finetuning with the pretraining task mixed in. They did not show whether doing the same scheduling mixing pretraining tasks with vanilla finetuning tasks would have a similar effect, or whether multi-task parallel training with both the pretraining task and the vanilla finetuning tasks would have a similar effect. This is a very important baseline that is completely missing. It is possible that a much simpler approach - simply mixing in some pretraining data/task at finetuning stage would have all the same effect.

3) The proposed formulation of the "unified data" seems narrow and classification heavy. It may not capture well generation heavy LLM use cases such as "write me a poem" that are very important and popular these days.

4) The evals are too narrow. For comparison with GPT3 and InstructGPT, the author should test few-shot and zero-shot performances on unseen tasks.

**Questions:**

1, Could the author please compare the proposed method to the following methods?
    1.1 Use the vanilla finetuning objective but the same scheduling of pretraining and finetuning, with and without reformatting proposed by the author.
    1.2 Use a mixture of the vanilla finetuning objective and the pretraining objective, with and without reformatting proposed by the author.

2, Could the author please focus the eval on out-of-domain tasks where the training split, or a training split of a similar task, is not available to the model, preferably in a few-shot / zero-shot setting?

---

### Meta-Review · Area_Chair_qNVQ · 2023-11-25

**Metareview:**

The authors propose an approach, FreeLM, to mixes various supervised finetuning (SFT) data with the pretraining data and pretrain a decode-only language model. FreeLM requires fitting existing SFT data from various tasks into a new binary classification format, where the supervision can be added as an extra term in the loss function with pretraining data. In language model training, pretraining and modified finetuning is conducted in iterations. The authors claim FreeLM eliminates the need for subsequent fine-tuning, and show FreeLM outperforms larger models like GPT-3 and InstructGPT on the GLUE benchmark at zero/few shot setting.

Strength:
 - The paper studies important and novel areas - language model training and unification of  pretraining and finetuning stages for more efficient and cleaner setting.
- The paper also studies whether the auto-regressive language modeling abilities has been impacted by adding SFT signals, and find that with their method this impact has been minimal. This is an encouraging sign.

Weakness:
 - The paper over claims and doesn't design experiment properly. FreeLM is claimed as "finetuning free" and compared with GPT-3 and InstructGPT at zero-shot and few-shot setting. However, training split data of eval tasks (or of similar eval tasks) are used in pretraining. Thus, FreeLM is trained on much more information (and similar domains to eval tasks) than baselines, which make the comparison not fair. This also leads to questions how generalizable to unseen domains/tasks FreeLM will be.
 - The method is not novel since the approach is simply a mixture of pretaining and multi-task finetuning.  More experiments and analysis are required to show how FreeLM surpasses previous techniques or introduces new capability to language models.
- FreeLM is limited to classification tasks. It's not clear how this approace can be extended to generative tasks or others. The introduction of SFT signals seems throwing away the main benefit of language models.

**Justification For Why Not Higher Score:**

As I summarized above in the weakness, I think this paper is lack of novelty and over claims. Both make the paper of limited interest to ICLR community and thus I recommend reject.

**Justification For Why Not Lower Score:**

N/A

---

### Decision · Program_Chairs · 2024-01-16

Reject